# Differences in Expression of *IQSEC2* Transcript Isoforms in Male and Female Cases with Loss of Function Variants and Neurodevelopmental Disorder

**DOI:** 10.3390/ijms23169480

**Published:** 2022-08-22

**Authors:** Beatriz Baladron, Lidia M. Mielu, Estrella López-Martín, Maria J. Barrero, Lidia Lopez, Jose I. Alvarado, Sara Monzón, Sarai Varona, Isabel Cuesta, Rosario Cazorla, Julián Lara, Gemma Iglesias, Enriqueta Román, Purificación Ros, Gema Gomez-Mariano, Isabel Cubillo, Esther Hernandez-San Miguel, Daniel Rivera, Javier Alonso, Eva Bermejo-Sánchez, Manuel Posada, Beatriz Martínez-Delgado

**Affiliations:** 1Instituto de Investigación de Enfermedades Raras (IIER), Instituto de Salud Carlos III (ISCIII), 28220 Madrid, Spain; 2Bioinformatics Unit, Instituto de Salud Carlos III (ISCIII), 28220 Madrid, Spain; 3Neuropediatrics Service, Hospital Puerta de Hierro, 28222 Madrid, Spain; 4Centro de Investigación Biomédica en Red de Enfermedades Raras (CIBERER), U758, 28029 Madrid, Spain

**Keywords:** *IQSEC2* gene, neurodevelopment syndrome, exome, transcript isoforms, intron variant, gene expression, SpainUDP

## Abstract

Pathogenic hemizygous or heterozygous mutations in the *IQSEC2* gene cause X-linked intellectual developmental disorder-1 (XLID1), characterized by a variable phenotype including developmental delay, intellectual disability, epilepsy, hypotonia, autism, microcephaly and stereotypies. It affects both males and females typically through loss of function in males and haploinsufficiency in heterozygous females. Females are generally less affected than males. Two novel unrelated cases, one male and one female, with de novo *IQSEC2* variants were detected by trio-based whole exome sequencing. The female case had a previously undescribed frameshift mutation (NM_001111125:c.3300dup; p.Met1101Tyrfs*5), and the male showed an intronic variant in intron 6, with a previously unknown effect (NM_001111125:c.2459+21C>T). *IQSEC2* gene expression study revealed that this intronic variant created an alternative donor splicing site and an aberrant product, with the inclusion of 19bp, confirming the pathogenic effect of the intron variant. Moreover, a strong reduction in the expression of the long, but also the short *IQSEC2* isoforms, was detected in the male correlating with a more severe phenotype, while the female case showed no decreased expression of the short isoform, and milder effects of the disease. This suggests that the abnormal expression levels of the different *IQSEC2* transcripts could be implicated in the severity of disease manifestations.

## 1. Introduction

The IQ motif and SEC7 domain containing protein 2 is coded by an X-chromosome gene (*IQSEC2)*, abundantly expressed in the brain which plays an important role in neuronal development, excitatory synapses and synaptic plasticity [1]. Mutations in *IQSEC2* gene cause X-linked intellectual developmental disorder-1, XLID1, (MIM: #309530), preferentially characterized by moderate to profound intellectual disability [2], often accompanied by seizures and autism spectrum disorder (ASD) [1,3,4]. A number of different pathogenic variants were identified by the exome sequencing of nonsyndromic intellectual disability [5]. There are overall more than 70 different pathogenic variants which include splice site, nonsense, missense, deletions, duplications and structural variants being responsible for associated phenotypes, with the majority of cases appearing de novo in the patients [6]. The genotype–phenotype correlation is not yet well-known, however, missense variants generally have a less severe phenotype than truncating variants [7]. Recent studies show that although *IQSEC2* mutations affect both sexes, females are variably and generally less affected than males [3,8].

The majority of genes located in the X chromosome are subjected to X chromosome inactivation (XCI) in human females. As a consequence, females carrying a pathogenic mutation in the X chromosome do not manifest or have milder symptoms of the disease while males show the affected phenotype. However, *IQSEC2* is one of approximately 20% of X chromosomal genes that escape XCI and therefore it is expressed from both X alleles in females. This phenomenon is largely unknown and complex, causing variable phenotypes in females with X-linked neurodevelopmental disorders [3,9].

Different isoforms have been described for the *IQSEC2* gene, resulting from alternative promoter usage and splicing. The largest isoform (NM_001111125, ENST00000396435.8) is 6037 bp in length and includes 15 exons which encode for a 1488 amino acid (aa) protein that is regulated by the upstream promoter (P2) and it is the most prominent isoform, expressed predominantly in the brain. An alternative downstream promoter (P1) regulates at least two other isoforms. The shortest isoform (NM_001243197, ENST00000639161.1) includes three alternative exons, which produce a 922 bp mRNA encoding a 73 aa protein. Thus, this isoform has a different coding region with distinct 5′UTR, N- and C-termini compared to the large isoform. Even though very little is yet known about this short protein, although its highly conserved sequence suggests an important role. In addition, another intermediate isoform (NM_015075, ENST00000375365.2), which undergoes alternative splicing, combines exons of both the short and large isoforms, producing an intermediate-sized transcript of 5208 bp that is translated into a 949 aa protein [4]. Both large and short isoforms are expressed across different body tissues, but intermediate isoforms are much lower expressed. The most highly expressed isoform in brain and in most tissues is the larger isoform but the short isoform also shows a significant expression in brain and skeletal muscle (Figure 1).

*IQSEC2* encodes the IQSEC2 protein, also known as BRAG1 (brefeldin A resistant Arf-guanine nucleotide exchange factor 1), a guanine nucleotide exchange factor (GEF) present in the postsynaptic density of synapses (PSD) in the forebrain. The principal substrate of IQSEC2 is the ADP-ribosylating factor 6 (ARF6) which regulates actin dynamics and membrane trafficking in neurons. [10]. BRAG1 is one of the most abundant proteins in the post-synaptic density of glutaminergic neurons, its role is crucial in post-synaptic transmission [11]. Related to its function, the gene presents highly conserved regions. At the protein level, there are several domains: IQ-like domain (amino acids 347–376) is a calcium calmodulin binding motif, SEC7 domain (aa 746–939) is responsible for guanine nucleotide exchange factor (GEF) [12], N-terminal coiled coil (CC) domain (aa 23–74) seem to promote self-assembly, pleckstrin homology (PH) domain (aa 951–1085) (aa 347–376) binds to phosphoionositides, and proline-rich motif (PRM) and PDZ binding motif (aa 1484–1488), required for interaction with synaptic PDZ proteins (PSD-95, SAP102, MAGI1, MAGI2) [13]. While the shortest IQSEC2 isoform (NM_001243197) encodes a protein that does not contain any of these domains, the intermediate isoform (NM_015075) is translated into a 949 aa protein that contains three of the functional domains of the longer form, the IQ-like motif (aa 142–174), the SEC7 domain (aa 541–735), the PH domain (aa 746–880) and lacks the CC and PDZ binding motif.

The Spanish Undiagnosed Rare Diseases Program, SpainUDP, aims to find a diagnosis for patients with unsolved rare diseases by applying genomic analysis together with deep phenotyping [14] and is a member of the UDNI—Undiagnosed Diseases Network International (https://www.udninternational.org/, accessed on 28 July 2022) [15]. Here, we present two cases, a male and a female with pathogenic IQSEC2 variants that cause the loss of functional protein. The expression analysis of the IQSEC2 gene allowed us to establish the pathogenicity of the intronic variant as the cause of the disease. Furthermore, quantitative expression analysis revealed different expression levels of IQSEC2 isoforms in both cases. Only the variant found in the male patient correlates with a pronounced reduction in both the large and short isoforms expression, suggesting that, in addition to the large isoform, the shortest IQSEC2 isoform could also influence the phenotypic manifestations of the IQSEC2-related disease and suggest an important role in brain function.

## 2. Results

### 2.1. Variants in IQSEC2 Gene in the Patients

The ND0673 male patient exhibits a severe neurological phenotype including psychomotor delay and hypotonia, detected at approximately 8 months of age, with epileptic spasms appearing at 5 years of age. Simultaneously, to the onset of epileptic encephalopathy, intellectual disability and autistic traits, he suffered severe neurologic regression, including a decrease in motor skills with ataxia, spasticity and gait difficulties. 

The ND1443 female patient has a less severe phenotype, with a neonatal normal period followed by global developmental delay. Currently, she has a neurodevelopmental disorder with behavioral problems, autistic features and language impairment.

Description of the phenotype of both cases is included in Table 1.

Exome sequencing of both families (ND067 and ND144) identified two potentially damaging loss of function variants in the IQSEC2 gene occurring de novo in probands. None of these variants had been previously reported in either ClinVar or gnomAD databases. The constraint score provided by the gnomAD based on the ratio of the observed/expected (o/e) number of loss of function variants in a gene, which shows for the IQSEC2 gene, a very low o/e ratio of 0.03 with a CI (0.01–0.13). This indicates that IQSEC2 has a high probability of being loss of function intolerant.

The male patient ND0673 was found to have an intronic variant at the genomic coordinate ChrX:g.53277882G>A (GRCh37), which corresponds to a hemizygous substitution C>T at the position +21 within intron 6 (NM_001111125:c.2459+21C>T). We suspected that this variant might interfere with the proper slicing of intron 6. The amplification of IQSEC2 mRNA with primers spanning exons 6–7 (Figure 1A) showed a higher band of 247 bp in the patient, while the parents showed the expected wild-type size product of 228 bp (Figure 2). The same aberrant band was observed in mRNA from muscular tissue sample of the patient. Sanger sequencing of the upper band revealed the inclusion of the first 19 bp of intron 6 in the mRNA product, confirming the use of an alternative donor splicing site created by the mutation in the patient (Figure 2). This aberrant splicing and the inclusion of the 19 nucleotides is predicted to produce a change in the open reading frame leading to the early protein termination of the IQSEC2 protein (p.Pro745Valfs*12). This specific variant was not found in gnomAD, but a similar variant affecting the same position but changing to adenine instead of thymine (c.2459+21C>A) is reported in gnomAD, but with a very low frequency of 0.00007379 in Europeans. 

The female patient ND1443, was found to have a heterozygous de novo variant consisting on a single adenine insertion, ChrX:g.53265654dup (GRCh37), located in exon 13 (NM_001111125:c.3300dup). The mutation produces a frameshift leading to the early termination of protein translation (p.Leu1027Profs*79).

### 2.2. Expression of IQSEC2 Long and Short Transcript Isoforms by QT-PCR

To determine the impact of the identified variants in the expression of the different IQSEC2 isoforms, we separately quantified the level of long isoforms (NM_001111125) and the short isoform (NM_001243197). For the analysis of the long isoforms expression, primers in two different regions were used, one at the proximal part (exons 2–3) and another one at a more terminal region (exons 12–13) (Figure 1A). The primers at exons 2–3 specifically detected the expression of the long isoform since exon 2 is only used in this isoform. However, the primers amplifying between exons 12 and 13 could detect the expression of both long and other intermediate isoforms (NM_015075). For the analysis of the shorter isoform, primers between alternative exons 2′ and 3′ allowed the detection of the short transcript in a specific way (Figure 1A). Both patients, male and female, showed a remarkable reduction in the expression of the long isoforms compared to their parents (Figure 3). The decrease was more evident in the male (ND0673), showing a 50–70% reduction in the expression, while the reduction was between 30 and 50% in the female patient (ND1443) compared to the parents. In both families, the progenitors expressed very similar levels of IQSEC2 long isoform. Regarding the short IQSEC2 transcript, a highly reduced expression was evidenced in the male patient, while no decrease was shown in the female patient (Figure 3).

## 3. Discussion

Two novel *IQSEC2* de novo truncating mutations were identified through whole exome sequencing in two patients from the Spanish Undiagnosed Rare Diseases Program, SpainUDP: an intronic variant in the ND0673 male patient (c.2459+21C>T) and a frameshift variant in the ND1443 female patient (p.Leu1027Profs*79), increasing the list of pathogenic mutations associated with this X-linked neurodevelopment syndrome [6]. It is described that neurological alterations in patients with *IQSEC2* pathogenic mutations used to be more severe in affected males than females. In accordance, the phenotype shown in our patients revealed much worse clinical manifestations in the male than in the female in terms of motor deficiency, intellectual disability and epilepsy. In this regard, previously described cases with the same mutation showed more severe developmental delay in males, providing evidence for sex differences in the severity of the disease [11].

The de novo intronic *IQSEC2* variant (c.2459+21C>T) found in the ND0673 male patient has not been previously described. It creates a new splice donor site, producing an aberrant transcript leading to a frameshift and premature protein termination. Some other splicing variants have been described in *IQSEC2*, although these types of variants are less frequent than other loss of function variants [6]. In some cases, splicing variants can modulate the associated phenotype, depending on the percentage of cells showing the abnormal splicing event [16]. In our patient, the expression analysis only detected the altered splicing pattern, suggesting that the novel donor splicing site created by the mutation is preferentially used, producing almost only aberrant transcripts. These altered transcripts are predicted to produce a truncated protein lacking part of the Sec7, PH and PDZ domains of IQSEC2. The absent expression of a normal transcript would explain the severe phenotype of the patient. In this study, the two mutations found were the loss of function variants, a frameshift and a splicing mutation. These variants either lead to the degradation of the mRNA by the nonsense-mediated decay mechanism or produce the early termination of the protein, mainly leading to its degradation.

The genotype–phenotype relationship for pathogenic variants in *IQSEC2* remains complex. It has been suggested that the phenotype is influenced not only by the variant effect, but also by the sex of the patient and by the protein-affected domain. It is described that patients harboring de novo *IQSEC2* mutations disrupting of the C-terminus of the IQSEC2 protein relate a mild phenotype [17], and that missense variants cause a less severe phenotype than truncating variants [7]. Missense pathogenic variants have been described to affect the three IQSEC2 functional domains (IQ-like, Sec7 and PH domain) [3,18]. In general, males with missense variants present mild–severe intellectual disability with variable penetrance of seizures and ASD traits, and less frequently with speech deficits, whereas most females with missense variants in heterozygosity are asymptomatic, or mildly affected with intellectual disability or learning difficulties [6]. Truncating variants, which are associated with more severe neurodevelopmental phenotype due to the loss of IQSEC2 function [7], are generally better tolerated in females than in males [3]. Male patients with loss of function variants generally present severe intellectual disability, seizures and speech deficiency [3,19,20], while female patients with loss of function variants in heterozygosity present a variability in phenotype severity for intellectual disability, seizures and global developmental delay [3,21]. 

Both the type and position of the mutation within the *IQSEC2* gene is relevant for the phenotype. According to a previous study [3], missense variants are more frequent in males than in females, probably because truncation variants may not allow protein functionality, while the missense variant in females is less pathogenic than the truncation variant because of the normal allele compensation. In the case of the ND0673 male, the intronic insertion would produce a truncated protein at the Sec7 domain. In the ND1443 female patient, the frameshift variant affects the more terminal PH domain, and shows a mildly affected neurological phenotype but more behavioral problems. The observation of females with frameshift variants leading to a late C-terminal protein truncation revealed that the four last amino acids, necessary for the interaction with PDZ proteins, are required for the function of the protein at the synapse [3]. Despite this, there is also a compensatory mechanism by the expression of the other allele in females that influences the less severe phenotype. The IQ-like and Sec7 functional domains are both involved in the catalytic activity of guanine nucleotide exchange and the activation of the substrate ARF6, and the PH domain is involved in the IQSEC2–accessory protein interactions [22]. The IQ-like and also Sec7 domains are both important in protein function [18] and their absence relates to phenotype severity, as we observed in the male patient. 

Regarding females, the less clinically affected phenotype is influenced by the presence of a normal X chromosome [8]. Nevertheless, studies of *IQSEC2* variants in females have led to undetermined results in terms of phenotype severity [11] and have to be interpreted keeping in mind that *IQSEC2* is a gene that escapes XCI in humans [23]. Some genes not subjected to X-inactivation result in differences in dosage between males and females, however, the *IQSEC2* expression, is globally similar in males and females, as we observed in our healthy parents [9,24,25]. The expression observed in ND1443 patient showed a decrease in the large *IQSEC2* expression, but the expression was higher than in the male patient, likely because of the expression of the other X chromosome. The similar expression of both the father and mother samples indicates that although the *IQSEC2* gene escapes to XCI, there must be a compensatory mechanism to ensure an accurate IQSEC2 dosage. The XCI is a complex process in females, which occurs at both the tissue and cellular levels [3]. Recent analysis of the XCI gene status evidenced that there is a substantial number of genes that present a variable inactivation status including an incomplete or skewed inactivation, suggesting an important correlation of the chromosomal region and inactivation pattern [6]. Altogether, data suggest alternative regulatory mechanisms of *IQSEC2* dosage probably determined by epigenetic modifications, modifier genes or skewed X inactivation which contribute to the wide clinical variability [17]. 

Moreover, different transcripts have been described by the *IQSEC2* gene by the alternative splicing of different exons and the use of an alternative promoter. Although the *IQSEC2* protein is mainly expressed in the nervous system (fetal and adult human brain and frontal cortex), there are differences in the expression levels of *IQSEC2* transcript isoforms across tissues. The long (NM_001111125) followed by the short isoform (NM_001243197) are the two more abundantly expressed isoforms in the brain; however, the short isoform is the main isoform expressed in muscle. The implication that different levels of the isoforms might have in the disease manifestations is still unknown and should be explored in more detail. Our quantitative analysis of the expression in the whole blood of the long isoform and the shortest isoform in the two patients reveal differences in the male and female patients. In addition to the expected decrease in the long *IQSEC2* isoform, with a more pronounced reduction in the male than in female, we almost found an absence of expression of the shortest isoform in the male but not in the female. This suggests that the intronic variant found in patient has an impact in the expression of the short isoform. The mechanism by which the variant affects the expression of this isoform is unknown. A potential mechanism that would explain this finding is that variant c.2459+21C>T interferes with the biding of a transcription factor. The preliminary analysis of neuronal transcription factors whose binding might be affected by the variant shows that the wild type sequence is a predicted binding site for the transcription factor ASCL1 that would be disrupted by the presence of the variant. Interestingly, the c.2459+21C>T variant would introduce a binding site for neural transcription factor SP9 (Figure 4). Changes in the transcription factor affinities for their biding sites might have an impact in the expression of the surrounding short variant. However, further work will be required to test this hypothesis. 

It is also unknown whether the lack of expression of the short isoform contributes to the neurological symptoms of the patient. In this sense, a female patient was previously described with an *IQSEC2* deletion, causing a frameshift located in the alternative exon 1 affecting only the short isoform, and presented neurological alterations [3]. Altogether, most pathogenic variants described in *IQSEC2* lead to loss-of-function of the longest isoform in hemizygous males and haploinsufficiency of the same isoform in heterozygous females. However, the loss of the short isoform at least in some patients suggests its likely involvement in the disease. The Regulation of the expression of the different *IQSEC2* transcripts in addition to other mechanisms other than XCI may determine *IQSEC2* dosage in females, contributing to the phenotype.

To conclude, we identified the two previously undescribed pathogenic loss of function variants in *IQSEC2* in a male and in a female patient. The quantification of the expression levels of the long and short transcripts isoforms revealed differences in these two cases. A higher decrease in the expression of the long isoform in the male was found compared to the female case, and although both variants occur downstream of the shortest isoform, the splicing variant in intron 6 in the male correlates with a strong reduction in short isoform expression and a more severe phenotype. Our results suggest the possible involvement of the *IQSEC2* short isoform in disease manifestations and should be further studied in more detail in additional patients. 

## 4. Materials and Methods

### 4.1. Patients

Patients, one male (ND0673) and one female (ND1443), were recruited by the SpainUDP [14] at the Institute of Rare Diseases Research (IIER), Institute of Health Carlos III (ISCIII). Both of them presented different neurodevelopmental problems and remained undiagnosed. Clinical characteristics of these two cases are shown in Table 1. Peripheral blood samples were collected from the patients and their parents to perform trio-based whole-exome sequencing. Informed written consent was obtained from all participants or their legal representatives. This study was approved by the Research Ethics Committee at the ISCIII.

### 4.2. Trio Exome Analyses

Genomic DNA was extracted using Qiagen QIAamp DNA kit. Trio-based Whole Exome Sequencing (WES) libraries were prepared using Nimblegen MedExome + ChrMit using HS2000 v4, 2x100 bp, in the ND067 family and Nextera Flex Enrichment kit (Illumina), and sequencing was carried out in a NextSeq 500 Illumina sequencer using High Output Flow Cell 2x75 cartridges (Illumina, San Diego, CA, USA) in the ND144 family. The DNA quality and concentration of the pre- and post-enriched libraries were assessed using the Agilent Technology 2100 Bioanalyzer and the Quantifluor system (Promega, Madison, WI, USA). The quality of raw output reads was first assessed using FastQC software v.0.11.3. Next, low-quality bases were filtered and trimmed with Trimmomatic v.0.33. This tool scans sequence reads and removes the 3′ ends where the average quality of bases within the specified window size falls below the selected quality threshold. We defined a sliding window size of 4 and an average quality of 15. Reads shorter than 70 bases in length were excluded from further analyses. A minimum of 86% of target exome regions were sequenced at read depths of at least ×20, whilst mean read depths were consistently above ×75 in all cases. After preprocessing, sequences were aligned to the human reference genome “Human_g1k_v37” using BWA v0.7.12, and PCR duplicate reads were marked with Picard v1.140. Local realignment of reads, quality recalibration and variant calling were jointly performed for each family trio using the Genome Analysis Toolkit HaplotypeCaller (GATK v3.4), in line with GATK best practices [26,27]. KGGSeq (v0.8) was used to filter potentially disease-causing variants in the proband based on the inheritance mode and genotype (i.e., homozygously or heterozygously acquired de novo/inherited in compound form), and to annotate variants with information on the affected gene, alternative transcripts, functional predictors, reported allele frequencies in population databases (MAF < 0.01), associated diseases and related bibliography. Intronic variants were checked using the Human Splicing Finder tool to identify potential effects or changes on RNA splicing. Lastly, we carefully inspected all the available scientific evidence of associations between the candidate variants and the diseases of interest through detailed research in public genetic databases (ExAc, GeneCard, NCBI, UniProt, OMIM, Pubmed, etc.). Sanger sequencing was performed to validate all candidate variants. 

### 4.3. IQSEC2 Gene Expression and Isoform Quantification

Different *IQSEC2* expression analysis approaches were designed to evaluate the effect of the intronic variant found in the male case on one hand. On the other hand, we quantitatively analyzed the large and short *IQSEC2* transcript isoforms.

The total RNA from blood leukocytes of all family members was isolated using the Speedtools Total RNA extraction kit (Biotools). In addition, RNA was also extracted from muscular biopsy obtained from theND0673 male patient. Then, RNA was reverse transcribed by Maxima First Strand cDNA Synthesis kit (Thermoscientific). The total RNA input for RT-PCR was 1 µg. To check the effect of the intronic variant located in intron 6 identified in the male case (ND0673), the obtained cDNA was amplified using a pair of primers designed to span the *IQSEC2* gene transcripts at the exons 6–7 junction (*IQSEC2*_Fwd: 5′GTGGGAGTGGCTCACTTCAT3′ and *IQSEC2*_Rv: 5′AGACTCATCGAAGCCTTCAGC3′). PCR products were subsequently loaded on a 2% agarose gel and electrophoresed to confirm the expected amplicon size (228 bp). Finally, target bands were excised, purified using QIAquick Gel Extraction Kit (Qiagen) and sequenced to check for mutation-induced effects on the splicing pattern in the patient.

Moreover, quantitative real-time PCR (QT-PCR) assays were designed and performed on both families to investigate whether the detected variants caused changes in the levels of expression of the different *IQSEC2* transcript isoforms in the patients relative to their parents. Commercially available Taqman probes (ThermoFisher Scientific, Waltham, MA, USA) that could discriminate between different *IQSEC2* isoforms were obtained. Probe Hs00323052_m1, which binds at the exons 2–3 junction, was used to detect the presence of the longest isoform (NM_001111125), and probe Hs06089735_m1, recognizing the alternative exons 2′-3′ boundary, was used for the shortest isoform (NM_001243197) (Figure 1A). We also used probe Hs00390333_m1, which binds both long and intermediate isoforms (NM_015075) at the shared exons 15–16 boundary. All assays were run on the QuantStudioTM 3 Real-Time PCR System (ThermoFisher, Waltham, MA, USA).

### 4.4. In Silico Prediction of Transcription Factor Binding

Transcription factors binding predictions from the JASPAR2022_CORE_vertebrates_non-redundant_v2 collection were analyzed using the Tomtom tool [28] in the MEME suite [29].

## Figures and Tables

**Figure 1 ijms-23-09480-f001:**
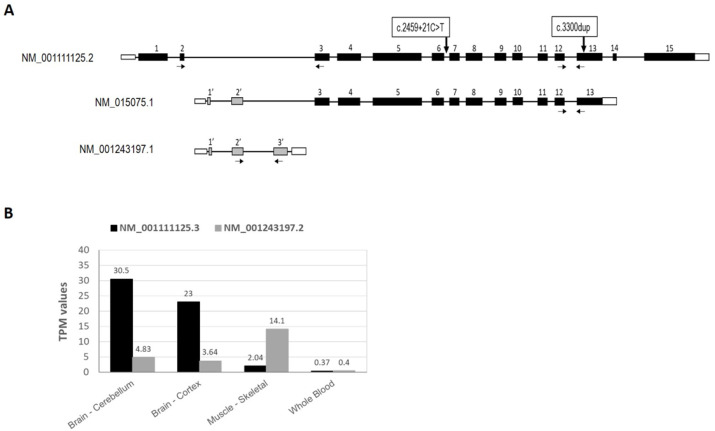
(**A**) Representation of IQSEC2 isoforms. Schematic representation of large isoform (NM_001111125) which include 15 exons; intermediate isoform (NM_015075) which include 14 exons; and short isoform (NM_001243197) which include 3 alternative exons. Primers used for the quantitative expression of the long and short isoforms are represented by arrows. (**B**) Isoforms expression of *IQSEC2* in different tissues obtained from GTEX database (https://www.gtexportal.org, accessed on 30 May 2022). TPM values corresponding to the larger *IQSEC2* isoform (NM_001111125) and the short isoform (NM_001243197) are represented as the expression values in brain cerebellum and cortex, skeletal muscle and whole blood.

**Figure 2 ijms-23-09480-f002:**
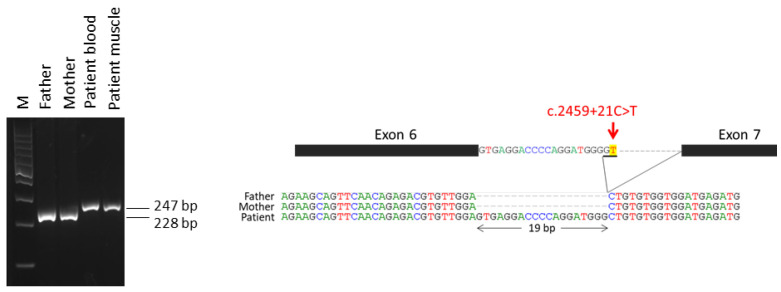
Variant effect on splicing in ND0673 patient. Conventional RT-PCR expression of *IQSEC2* mRNA in peripheral blood samples from the patient and his parents, and patient muscle sample. M: Molecular weight marker (100 bp DNA Ladder, Promega). Schematic representation of the mutation (c.2459+21C>T) in NM_001111125 transcript and the newly created donor slicing site. Sanger sequencing of aberrantly spliced products reveals the insertion of 19 bp of intron 6 in the patient.

**Figure 3 ijms-23-09480-f003:**
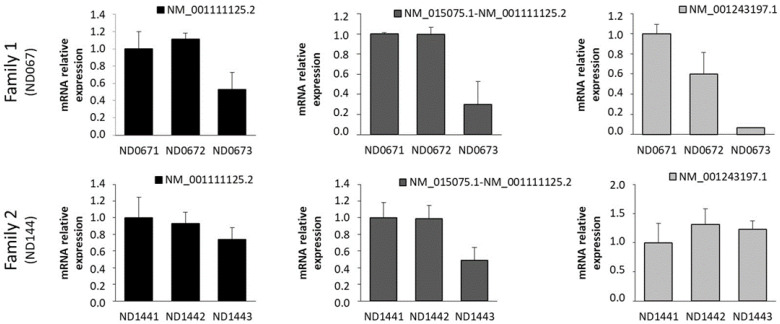
Relative quantification of IQSEC2 transcripts isoforms in the two families. Quantitative QT-PCR expression analysis for IQSEC2 isoforms in ND067 and ND144 families, showing expression levels of patients (ND0673 and ND1443) compared to their fathers (ND0671 and ND1441) and mothers (ND0672 and ND1442). Left panels represent expression levels based on primers amplifying between exons 2 and 3, specific for larger isoform (NM_001111125). Middle panel shows the expression using primers for exons 12–13, which do not discriminate between large and intermediate isoforms. Right panels show the expression of short IQSEC2 isoform (NM_001243197) with a specific assay designed between alternative exons 2′ and 3′, only present in this isoform.

**Figure 4 ijms-23-09480-f004:**
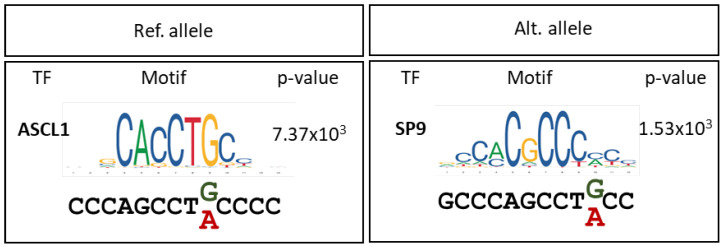
In silico prediction of transcription factor binding. Transcription factors with differential binding to variant c.2459+21C>T (Alt. allele) and the wild-type sequence (Ref. allele) are depicted. Binding motifs and *p*-values are also shown.

**Table 1 ijms-23-09480-t001:** Description of clinical characteristics of the patients.

*Patient*	*ND0673*	*ND1443*
**Variant (transcript)**	NM_001111125: c.2459+21C>T	NM_001111125:3300dup
**Variant (protein)**	p.Pro745Valfs*12	p.Leu1027Profs*79
**Mutation effect**	Splicing	Frameshift
**Inheritance**	De novo	De novo
**Patient’s gender**	Male	Female
**Age**	14	15
**Family history**	Brother and parents healthy	Maternal cousins with intellectual disability and Down syndrome
** *Clinical features* **		
**Neurological manifestations**	Intellectual disability	Intellectual disability
	Seizures	Focal-onset seizure
	Global developmental delay	
	Epileptic encephalopathy	
	Abnormal pyramidal signs	
	Cognitive impairment	
	Ataxia	
	Spasticity	
	Motor delay	
	Generalized hypotonia	
	Unsteady galt	
	Difficulty walking	
**Speech deficits**		Delay speech and language development
**Behavioral disturbances**	Autistic behaviour	Autistic behavior
		Obsessive-compulsive and aggressive behavior
		Stereotypy
		Hyperactivity
		Low frustration tolerance
		Impulsivity
**Metabolic test**	Normal	
**Brain magnetic resonance**	Normal in the first year	Normal
	Discrete myelination delay at 2 y/o	
**Computed axial tomography**		Normal
**Electroencephalogram**	Abundant epileptic anomalies especially in the bilateral frontal region at 5 y/o	Normal at 9 years of age
		Epileptic anomalies in the right anterior fronto-temporal region at 13 years of age
** *Other genetic studies* **		
**Kariotype**	-	Normal
**Array-CGH**	Normal	Normal
**Gene panel for epilepsy**	Negative	
**Mitochondrial DNA study**	Normal	

## Data Availability

Publicly available datasets regarding gene expression in different tissues by RNASeq were analyzed in this study. This data can be found in GTEx Portal: [https://www.gtexportal.org/home/, accessed on 30 May 2022]. The exome data results presented in this study are available on request from the corresponding author. The data are not publicly available due to privacy restrictions.

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
