# Peer review of "Differences in Expression of IQSEC2 Transcript Isoforms in Male and Female Cases with Loss of Function Variants and Neurodevelopmental Disorder"

_ijms, 2022, doi:10.3390/ijms23169480_

Round 1
Reviewer 1 Report
Only one major comments.
Are there other cases with similar symptoms? Could check if these patients have this novel variant. Could also compare the WES data of the male patient and his brother.
Author Response
First, we would like to thank the revision of the manuscript and the suggestions to improve it.
The symptoms shown by the two patients included in our study are common in other neurodevelopmental syndromes. This is why is very important to find the molecular alteration in specific genes to further try developing specific therapies.
None of the loss of function variants found in the two patients is reported in the gnomAD genomic database, which includes more than 100,000 exome sequences and more than 15,000 whole-genome sequences originated from disease and population based studies. The constraint score provided by gnomAD based on the ratio of the observed/expected (o/e) number of loss-of-function variants in this gene, shows for the IQSEC2 gene, a very low o/e ratio of 0.03 with a CI (0.01 - 0.13). This indicates that IQSEC2 shows a high probability of being loss-of-function intolerant. We have included this sentence in the manuscript, line 138.
The variant found in the male case, NM_001111125:c.2459+21C>T, has not been reported in gnomAD and it is not present in his parents. Thus, this is likely a de novo variant occurring in this patient with a very low probability to find other patients with the same variant. A similar variant is reported in gnomAD (c.2459+21C>A). This is also a very rare variant (with a frequency of 0.00007379 in Europeans), affecting the same position but changing to adenine instead of thymine and no phenotype described for it. We have added some of this information in the manuscript, line 153.
The patient’s brother was not analyzed. Blood sample to perform exome analysis was not available. However, since both parents do not have the variant, it is not likely that the brother with a typical phenotype had the variant found de novo in the patient.
Reviewer 2 Report
The manuscript by Baladron et al. reported two novel mutations in the IQSEC2 gene in patients with neurodevelopmental disorders. The authors characterized one frameshift mutation and an intronic variant in intron 6 affecting the mRNA expression of IQSEC2. The study is straightforward, and the manuscript is well-written. The interpretation is consistent with the findings. However, I have some questions related to the functional characterization of the mutations. Below are my concerns:
1. Why did the authors not check the effect of these mutations at the protein level? Is it possible to check the level of IQSEC2 protein in the family? Is it possible with the blood sample? Many times, due to incomplete NMD, there will be mRNA but no protein if checked by western blot.
2. If samples are not available, authors should try characterizing at least intronic mutations by in vitro analysis. There are many specific vectors available to characterize splice-site mutations. First, the authors should characterize if there is aberrant splicing due to this mutation. Second, create this intronic mutation (in IQSEC2) in a cell culture model system and check if there is a change in mRNA and protein levels of IQSEC2 due to this mutation in wild-type and mutant cell lines.
The manuscript has many typos and grammatical errors, I have listed some of them:
1. Line 91> calmodulim>> calmodulin
2. Line 93> aminoacids >> amino acids.
3. Line 95> phosphionositides>> phosphoionositides.
4. Line 107> stablish>> establish.
5. Line107-113> This single sentence has typos, punctuation, and grammatical errors. errors. Please modify this sentence carefully.
6. Table 1. Low frustation>> Low frustration.
7. Line 149> heterocigous>>I believe it should be written as heterozygous.
8. Line 176-178> Please rewrite this sentence.
A more tightly worded manuscript with the above-suggested corrections will make the paper a good read.
Author Response
Here we included the point-by-point response to the reviewer comments.
The manuscript by Baladron et al. reported two novel mutations in the IQSEC2 gene in patients with neurodevelopmental disorders. The authors characterized one frameshift mutation and an intronic variant in intron 6 affecting the mRNA expression of IQSEC2. The study is straightforward, and the manuscript is well-written. The interpretation is consistent with the findings. However, I have some questions related to the functional characterization of the mutations. Below are my concerns:
We thank very much to the reviewer for the revision and comments to improve the manuscript.
- Why did the authors not check the effect of these mutations at the protein level? Is it possible to check the level of IQSEC2 protein in the family? Is it possible with the blood sample? Many times, due to incomplete NMD, there will be mRNA but no protein if checked by western blot.
The expression of IQSEC2 gene is highly specific for the brain with some isoforms also expressed in skeletal muscle. Whole blood expresses only low mRNA levels, which can be detected by RT-QT-PCR, as demonstrated in our study.
The two mutations found are loss of function variants, a frameshift and a splicing mutation. These variants either lead to degradation of the mRNA by the nonsense mediated decay mechanism or produce the early termination of the protein, leading mainly to its degradation. So, no protein is typically expected. This fact, together with the low expression of the gene in blood samples prompted us not to check for protein.
We added a sentence in the manuscript, line 221, to underline the fact that loss of function variants usually lead to lack of protein production.
- If samples are not available, authors should try characterizing at least intronic mutations by in vitro analysis. There are many specific vectors available to characterize splice-site mutations. First, the authors should characterize if there is aberrant splicing due to this mutation. Second, create this intronic mutation (in IQSEC2) in a cell culture model system and check if there is a change in mRNA and protein levels of IQSEC2 due to this mutation in wild-type and mutant cell lines.
In our case, samples from the patient and the parents were available, so the splicing analysis was performed directly on their samples. We used blood sample and a muscle sample from the patient to evaluate the effect of the intronic variant on the splicing. As showed in the results, the patient had an altered splicing with a larger product partially including the intronic sequence.
In case any alteration was not detected, we agree that in vitro studies would had been appropriate to probe and characterize the aberrant splicing.
The manuscript has many typos and grammatical errors, I have listed some of them:
- Line 91> calmodulim>> calmodulin
It was changed in the text.
- Line 93> aminoacids >> amino acids.
It was changed in the text
- Line 95> phosphionositides>> phosphoionositides.
It was changed in the text
- Line 107> stablish>> establish.
It was changed in the text.
- Line107-113> This single sentence has typos, punctuation, and grammatical errors. Please modify this sentence carefully.
We have modified the sentence as follows:
“Expression analysis of IQSEC2 gene allowed us to establish the pathogenicity of the intronic variant as the cause of the disease. Furthermore, quantitative expression analysis revealed different expression levels of IQSEC2 isoforms in both cases. Only the variant found in the male patient correlates with a pronounced reduction of both the large and short isoforms expression, suggesting that in addition to the large isoform, the shortest IQSEC2 isoform could also influence phenotypic manifestations of the IQSEC2 related disease and suggest an important role in brain function.”
- Table 1. Low frustation>> Low frustration.
It was changed in the text.
- Line 149> heterocigous>>I believe it should be written as heterozygous.
It was changed in the text.
- Line 176-178> Please rewrite this sentence.
We changed the sentence in manuscript, line 184-186, to:
Regarding the short IQSEC2 transcript, highly reduced expression was evidenced in the male patient, while no decrease was shown in the female patient (Figure 3).
A more tightly worded manuscript with the above-suggested corrections will make the paper a good read.
We have reviewed the whole text thoroughly.